# Putting Data at the Centre of
# Offline Multi-Agent Reinforcement Learning

**Claude Formanek**                                                C.FORMANEK@INSTADEEP.COM
*University of Cape Town & InstaDeep, South Africa*

**Louise Beyers**                                                     L.BEYERS@INSTADEEP.COM
*InstaDeep, South Africa*

**Callum Rhys Tilbury**                                            C.TILBURY@INSTADEEP.COM
*InstaDeep, South Africa*

**Jonathan P. Shock**                                           JONATHAN.SHOCK@UCT.AC.ZA
*University of Cape Town, South Africa*

**Arnu Pretorius**                                                 A.PRETORIUS@INSTADEEP.COM
*InstaDeep, Rwanda*

## Abstract

Offline multi-agent reinforcement learning (MARL) is an exciting direction of research that uses static datasets to find optimal control policies for multi-agent systems. Though the field is by definition data-driven, efforts have thus far neglected data in their drive to achieve state-of-the-art results. We first substantiate this claim by surveying the literature, showing how the majority of works generate their own datasets without consistent methodology and provide sparse information about the characteristics of these datasets. We then show why neglecting the nature of the data is problematic, through salient examples of how tightly algorithmic performance is coupled to the dataset used, necessitating a common foundation for experiments in the field. In response, we take a big step towards improving data usage and data awareness in offline MARL, with three key contributions: (1) a clear guideline for generating novel datasets; (2) a standardisation of over 80 existing datasets, hosted in a publicly available repository, using a consistent storage format and easy-to-use API; and (3) a suite of analysis tools that allow us to understand these datasets better, aiding further development.

**Keywords:**   offline multi-agent reinforcement learning, offline reinforcement learning, multi-agent systems, reinforcement learning, datasets

## 1 Introduction

Many complex real-world problems can naturally be formulated as multi-agent systems—e.g. managing traffic (Zhang et al., 2019), controlling fleets of ride-sharing vehicles (Sykora et al., 2020) or a network of trains (Mohanty et al., 2020), optimising electricity grid usage (Khattar and Jin, 2022), and improving dynamic packet routing in satellite communication (Lozano-Cuadra et al., 2024). Improving on solutions to such problems is an important endeavour, because of the potentially immense societal benefits that they offer. Multi-Agent Reinforcement Learning (MARL) is a promising avenue to finding solutions to such problems, but the field faces a host of hurdles which must first be overcome. One key difficulty is the access to

accurate and efficient simulators, for online experience generation and exploration. To learn robust policies, extensive interactions with an environment are usually required (Yu, 2018), which makes simulator efficiency of paramount importance. Yet, for real-world applicability, the fidelity between the simulator and reality must also be maintained. Unfortunately, this balance of achieving high throughput in an online simulator while maintaining realistic dynamics is difficult, and practitioners must often rely on more basic environments with simplifying assumptions. The situation is particularly challenging when there are many agents interacting in complex ways, as is the case in MARL.

What typically does exist in such systems as those described above is the ability to capture large amounts of useful data. Across a range of complex control scenarios, even in situations where many agents are acting and the physical dynamics are not well understood (i.e. where designing a bespoke simulator would be very challenging), it may be straightforward to record data during operation. This opportunity is what *offline* RL leverages, by bridging the gap between RL and supervised learning. In the offline domain, the aim is to develop algorithms that use large, existing datasets of sequential decision-making transitions (whether recorded from the real-world, or created in simulation, or a mixture thereof) to learn optimal control policies, which can later be deployed online (Levine et al., 2020). The offline paradigm promises to help unlock the full potential of RL when applied to the real-world, where success has thus far been limited (Dulac-Arnold et al., 2021; Rafael Figueiredo Prudencio, 2024). In the multi-agent setting, algorithms are designed to learn a *joint* policy from a static dataset of previously collected multi-agent transitions, generated by a set of interacting behaviour policies.

Single-agent offline RL has enjoyed relatively widespread research attention and success (Prudencio et al., 2023). Core to such developments, the community has benefited greatly from standardised and publicly available datasets—as found in libraries such as D4RL (Fu et al., 2020) and RL Unplugged (Gulcehre et al., 2020). Yet, in the multi-agent case, such offerings are limited both in number and in quality. In fact, we argue in this paper that work in offline MARL has disproportionately focused on algorithmic innovation, and neglected the role of data almost entirely. It is common practice for authors to generate their own datasets for their experiments, almost as an after-thought, while little effort has been made to understand how the quality and content of these multi-agent datasets affect training dynamics and final performance. Such insights have been immensely valuable in the single-agent setting (Schweighofer et al., 2022), and it is known that multi-agent systems face additional complexities (Tilbury et al., 2024), where similar insights would be particularly useful.

Importantly, it should be clear that the end-goal for offline MARL is for systems to be deployed in the real world: real datasets, yielding real policies, useful in real applications. Yet, the path there is long and winding. To make progress, we focus in this paper on a simplified context: cooperative scenarios (where existing online MARL research is most mature, and many real-world examples exist), with data recorded from simulators. Solving these simpler problems does not magically solve the more complex ones, but we start here as a necessary first step towards deploying MARL in the wild. Notice that this journey parallels the one that computer vision has undertaken, as an example of a more established field of machine learning—before starting to tackle the enormously challenging task of, say, end-to-end learning from raw pixels for a self-driving car in the real-world (e.g. Bojarski

et al. (2016)), it was important to first tackle simpler problems, like handwritten digit recognition (LeCun et al., 1989). Focusing on fundamental datasets like MNIST (LeCun et al., 2010) has been integral to the progress in computer vision. We make similar efforts for offline MARL here.

Our paper is structured in the following way. We start surveying the current state of the field in Section 2, by studying how authors have, until now, been treating data in their research—with the evidence showing a general lack of care in the way data is considered. We build on this finding in Section 3 to show why this carelessness is problematic. Through four clear examples, we show how the specifics of the data has a significant impact in the learned performance of algorithms, something which has previously been overlooked. We respond in Section 4 with three contributions: firstly, a clear guideline on how data should be treated in offline MARL going forward; secondly, a standardised set of datasets, comprising over 80 `environment-scenario-quality` combinations, with a well-documented and accessible API, and an easy mechanism to extend this repository; and finally, useful tooling for researchers to understand the nature of their datasets, as an initial effort to promote data awareness in offline MARL. These contributions are all publicly available in our GitHub repository.[1]

## 2 The Current State of Datasets for Offline MARL

Offline MARL remains a relatively nascent field, with only a handful of papers released on the topic to date (see Table 1), but progress is accelerating. We want to understand how authors have been handling the data component of their research in the work done thus far by trying to answer the question: what is the state of the field, with respect to data itself? To do so, we present a comprehensive survey of work in empirical offline MARL, from leading peer-reviewed academic venues, to assess (1) how their data were generated and (2) what information about their datasets was provided. Though a simple assessment, we find these two axes already particularly telling in what they reveal. Table 1 summarises our findings.

From Table 1, we firstly notice that the majority of papers assessed generated their own datasets. Each paper also creates these datasets in different ways—using a wide variety of underlying online algorithms to learn policies for the generated trajectories. The dataset labels themselves also vary across papers, with no consistent naming convention. Information about the dataset properties is also sparse. To measure a given dataset's 'quality', the mean episode return of trajectories is often reported, but even this metric is not always given. The return distribution is mostly ignored, except for the occasional proxy of reporting the standard deviation. Essentially, there is little information presented on the contents and diversity of the experience in a given dataset. Yet these aspects of a dataset are crucial to the resulting performance of offline learning. In the single-agent literature, it has been shown that dataset properties have a marked impact on results (Schweighofer et al., 2022). In the multi-agent context, other complex aspects of coordination (Tilbury et al., 2024; Barde et al., 2024) make the contents and characteristics of datasets even more important to understand.

We observe here that the field of offline MARL has struggled to find common ground to benchmark proposed algorithms. Even accepting that many authors generate their own datasets for their papers, there has been carelessness in reporting information about such datasets. Ultimately, the current reality points to a general lack of consideration around the

---

1. https://instadeepai.github.io/og-marl/

Table 1: A survey of the use of data in empirical offline MARL literature, focusing on (1.) where the data came from, and (2.) which properties of the data were reported. We notice that most of the assessed papers self-generated their data. We also see that there is no consistent naming strategy in the dataset qualities, and that the mean and standard deviation of the datasets are often absent.

| Paper | Source of Datasets | Dataset Labels | Means, $\mu$ | Standard Deviations |
|---|---|---|---|---|
| MAICQ (Yang et al., 2021) | **Self-generated** using DOP (Wang et al., 2020) | *good* / *medium* / *poor* | $15 < \mu < 20$ / $10 < \mu < 15$ / $0 < \mu < 10$ | **Not given** |
| MADT (Meng et al., 2021) | **Self-generated** using MAPPO (Yu et al., 2022b) | *replay* | Given | Given |
| Offline MARL with Knowledge distillation (Tseng et al., 2022) | Some **self-generated** using random policies or expert policies from PPO (Schulman et al., 2017), and some from MADT (Meng et al., 2021) | *good* / *normal* / *poor* | Given | Given |
| OMAR (Pan et al., 2022) | **Self-generated** using MATD3 (Ackermann et al., 2019) | *random* / *medium-replay* / *medium* / *expert* | **Not given** | **Not given** |
| CFCQL (Shao et al., 2023a) | **Self-generated** using QMIX (Rashid et al., 2018) | *medium* / *medium-replay* / *expert* / *mixed* | **Not given** | **Not given** |
| OMAC (Wang and Zhan, 2023) | Random Subsets from MADT (Meng et al., 2021) | *good* / *medium* / *poor* | **Not given** | **Not given** |
| OMIGA (Wang et al., 2023b) | Random Subsets from MADT (Meng et al., 2021) | *good* / *medium* / *poor* | Given | **Not given** |
| SIT Framework (Tian et al., 2023) | **Self-generated** using QMIX (Rashid et al., 2018) and FacMAC (Peng et al., 2021) | *low quality* / *medium quality* / *random quality* | Given | **Not given** |
| AlberDICE (Matsunaga et al., 2023) | **Self-generated** using MAT (Wen et al., 2022) | *expert* / *medium-expert* | **Not given** | **Not given** |
| Value Deviation & Transition Normalisation (Jiang and Lu, 2023) | Some **self-generated** using random policies or using QMIX (Rashid et al., 2018) or SAC (Haarnoja et al., 2018), and some decomposed from D4RL (Fu et al., 2020) | *random* / *medium* / *replay* / *expert* | **Not given** | **Not given** |
| State Augmentation via Self-Supervision (Wang et al., 2023a) | **Self-generated** using QMIX (Rashid et al., 2018) | *inadequate* / *moderate* / *superb* | $0 < \mu < 10$ / $10 < \mu < 15$ / $15 < \mu < 20$ | **Not given** |
| MADiff (Zhu et al., 2023) | Datasets from Formanek et al. (2023a) and from Pan et al. (2022) | *good* / *medium* / *poor* | Given | Given |
| MOMA-PPO (Barde et al., 2024) | **Self-generated** using MAPPO (Yu et al., 2022b), and decomposed from D4RL (Fu et al., 2020) | *random* / *medium* / *replay* / *expert* / *expert-mix* | Given | **Not given** |

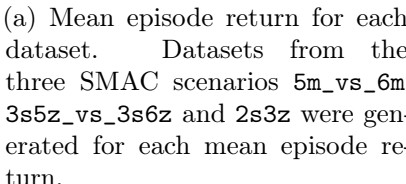

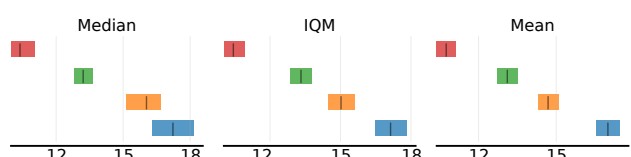

(a) Mean episode return for each dataset. Datasets from the three SMAC scenarios `5m_vs_6m`, `3s5z_vs_3s6z` and `2s3z` were generated for each mean episode return.

(b) A comparison of the performance obtained using datasets with different mean episode returns. Results are aggregated (across two algorithms, 10 random seeds, 32 evaluation episodes and three scenarios) using bootstrap confidence intervals (Agarwal et al., 2021).

Figure 1: To demonstrate the effect that the mean episode return of a dataset has on the final performance of an offline MARL algorithm, we generated four datasets with mean episode returns given in Figure 1a. We then train an offline MARL algorithm for 50k training steps on each of the datasets and compare the final performance of the algorithm across the different datasets. We repeat the experiment across three different SMAC scenarios, two different algorithms (`IQL+CQL` (Formanek et al., 2024) and `MAICQ` (Yang et al., 2021)) and 10 random seeds. The aggregated results are given in Figure 1b.

role of data in the field, where authors are failing to adequately control for the impact that data can have on experimental results. We will now show why a lack of such data-centrism is problematic for the field.

## 3 Why Dataset Characteristics Matter

Claims of algorithmic improvement become moot if a common basis of data is missing, since a dataset is one of the control variables in empirical offline MARL experiments which can impact performance significantly. In this section, we investigate why dataset characteristics significantly influence algorithmic performance in offline MARL. In subsection 3.1 we provide a series of illustrative experiments demonstrating how specific dataset properties, such as mean episode return, diversity, and underlying distributions, may affect algorithm outcomes, even when superficially similar metrics are presented. In subsection 3.2, we undertake a comprehensive correlation analysis spanning diverse datasets and algorithms drawn from the recent literature, shedding light on how varying dataset attributes interact with algorithm effectiveness. Ultimately, as summarized in the concluding takeaways, our findings underscore an urgent need for transparent, detailed reporting and standardization of datasets to ensure reproducibility, reliability, and meaningful advancement in offline MARL research.

### 3.1 Investigating the Effects of Dataset Properties on Algorithm Performance

**Dataset Mean.**    We begin our illustration with likely the most intuitive example: what happens to final performance when the average return of the dataset changes? We construct four distinct datasets with increasing means on three scenarios (`5m_vs_6m`, `3s5z_vs_3s6z`

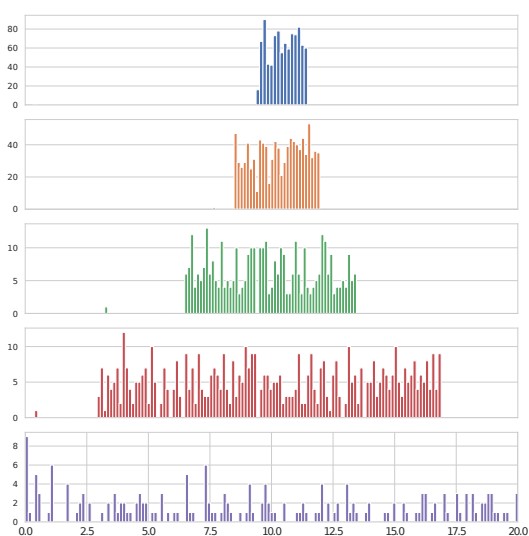

(a) Histograms of the episode returns of five datasets with the same mean but different standard deviations.

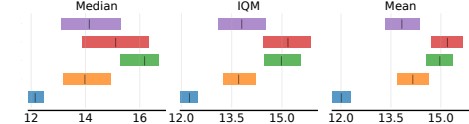

(b) Dataset mean and standard deviation.

(c) The aggregated results across scenarios (5m_vs_6m, 3s5z_vs_3s6z and 2s3z), algorithms (IQL+CQL and MAICQ) and random seeds are given for each dataset. The median, interquartile mean (IQM), and mean are all given with bootstrap confidence intervals (Agarwal et al., 2021).

Figure 2: To demonstrate the surprising effect that the standard deviation (std) can have on the performance of an offline MARL experiment we generate 5 datasets that each had the same mean but differing std. We then train two offline MARL algorithms, IQL+CQL and MAICQ, on the data and report the final performance. We repeat the experiment across three different SMAC scenarios, two different algorithms, 10 random seeds and an evaluation batch size of 32. We then aggregate the results as per Gorsane et al. (2022).

and 2s3z) from the SMACv1 environment (Samvelyan et al., 2019), by subsampling episodes from OG-MARL datasets (Formanek et al., 2023a). For each dataset, we fix the standard deviation at approximately 2.0 as calculated over 2000 episodes. We then train two offline MARL algorithms, IQL+CQL (Formanek et al., 2024) and MAICQ (Yang et al., 2021), on the individual datasets and report the final evaluation episode return, averaged across 10 random seeds. We present the aggregated results using bootstrap confidence intervals (Agarwal et al., 2021; Gorsane et al., 2022) in Figure 1.

We see that the aggregated final performance of IQL+CQL and MAICQ is positively correlated with the mean episode return of the dataset used for training. As the average quality of the data improves, so does the offline learning from this dataset—an intuitive result. This relationship is often used in the single-agent literature, where the final performance is reported as a percentage of the mean return in the dataset (Agarwal et al., 2019; Gulcehre et al., 2020), but doing so is rare in the multi-agent domain. In fact, the simple metric of the dataset mean is sometimes omitted entirely, as shown in Table 1.

**Dataset Spread.** Another important property of a dataset in offline RL is the diversity of the experience (Agarwal et al., 2019). In the single-agent setting, it is well-understood that many offline RL algorithms benefit from more diverse datasets (Schweighofer et al.,

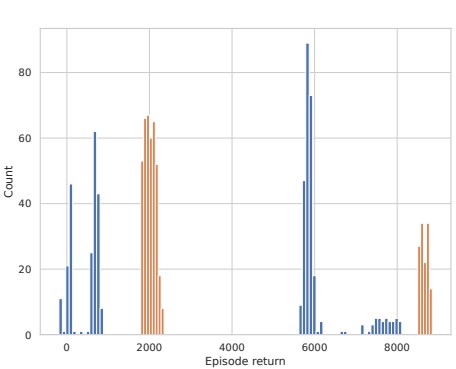

(a) Histogram of the episode returns for two datasets with similar means and standard deviations.

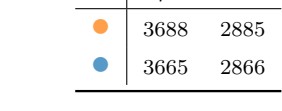

(b) Mean and standard deviation.

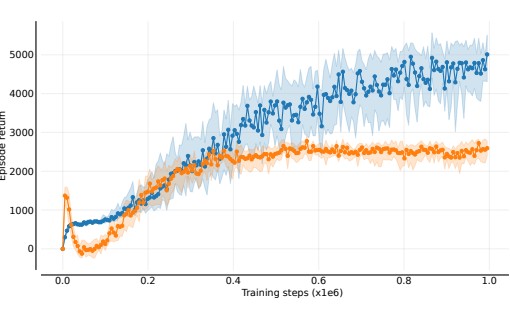

(c) Training curves.

Figure 3: We generate two datasets on 2-Agent Halfcheetah each with very similar episode return means and standard deviations, but distinct data distributions. We then train `MADDPG+CQL` on each dataset and report its performance over 1 million training steps. We repeat the experiment over 10 random seeds.

2022). Usually, this benefit arises because diverse experience leads to better coverage of the state and action spaces, resulting in fewer out-of-distribution actions, which are known to cause issues when training offline (Fujimoto et al., 2019). However, the effect of diversity in the multi-agent setting is less well understood and should not be taken for granted. For example, Tilbury et al. (2024) demonstrates a more complex relationship between dataset diversity and final performance in the offline MARL setting than might be initially assumed. In the literature, the standard deviation (std) of the data's episode returns is sometimes reported as a proxy for diversity.

We thus continue our illustration with another example: what happens when solely the std of the returns in a dataset changes? We now construct five distinct datasets, each with the same mean, but with increasing spread around that mean, keeping the number of included episodes constant. We create these datasets by subsampling from OG-MARL. Once again, we look at the three SMAC scenarios `5m_vs_6m`, `3s5z_vs_3s6z` and `2s3z`. Figure 2 shows the respective histograms of the datasets, and the corresponding aggregated results across two algorithms (`IQL+CQL` and `MAICQ`) and 10 random seeds.

In this situation, we begin to see a more complex relationship emerge. Rather than a simple linear relationship between diversity and performance, optimal results are found at intermediate levels of std. This result is less intuitive than before (where a higher mean return simply meant higher performance), and begins to hint at the impact of multi-agent dynamics. It is worth reiterating that very few papers report the std (or some other proxy for diversity) of their datasets—just three of the thirteen in Table 1 had done so.

**Dataset Distribution.** Our illustration continues with a question that builds on the previous two: controlling for equal mean and std, can two different dataset distributions yield different results? We construct two more datasets subsampled from OG-MARL with this

property.[2] We look at the `2halfcheetah` scenario from MAMuJoCo (Peng et al., 2021). We train `MADDPG` (Lowe et al., 2017) with `CQL` (Kumar et al., 2020) on these respective datasets, and show the results in Figure 3.

We can see here that an offline algorithm, when trained on two datasets with nearly identical summary statistics, can yield significantly different final performances. In essence, these metrics of a dataset only paint a limited view of the underlying distributions, yet these distributions may have a notable impact on the results. Here, we note that even those authors from Table 1 who have reported the mean and std of their datasets, have nonetheless omitted a visualisation of their datasets for further understanding. Therefore, authors might be missing key insights on the characteristics of the data they are using in their work.

**Dataset Coverage.** We conclude our illustration with possibly the most illuminating example of the subtleties of dataset charateristics and their effect on performance. We ask: can we have a situation where the return distributions are very similar, such that the summary statistics are similar and the histograms are closely aligned, yet yield significantly different results for the same offline algorithm?

For this example, we look at two publicly available datasets, from OG-MARL (Formanek et al., 2023a) and CFCQL (Shao et al., 2023a) respectively. We consider the `5m_vs_6m` scenario from SMACv1 (Samvelyan et al., 2019), and take the `Medium` quality dataset from each source. To further control the experiment, we subsample the original datasets to around $140k$ transitions, matching the distributions of episode returns in the process. The result is not only equal mean and std but also nearly identical, visually indistinguishable histograms, which we show in Figure 4a and detail in Table 4b. We train `IQL+CQL` on both datasets, across 10 random seeds. The results are illustrated in Figure 4c.

Table 2: The Joint State-Action Coverage (Joint-SACo) scores for two datasets on the `5m_vs_6m` SMAC task, one from OG-MARL and the other from CFCQL (Shao et al., 2023a). A lower score means there is less Joint State-Action coverage in the dataset, i.e. there are more repeated transitions in the dataset.

| Dataset | Quality | Joint-SACo score |
|---|---|---|
| ● CFCQL | `Medium` | 0.10 |
| ● OG-MARL | `Medium` | 0.83 |

We see here a curious outcome—despite the return distributions being essentially the same, the achieved algorithm performance is significantly different. The difference cannot be explained solely by the statistics and distribution of the episode returns. Evidently, there are other significant differences between the datasets which escape the reach of our current lens. We note that episode return is itself a summary of a trajectory, and is an abstraction of the actual experience in the dataset.

To better understand what is happening here, we extend to the multi-agent setting the approach from Schweighofer et al. (2022) to measure dataset diversity. Specifically, we extend their State-Action Coverage (SACo) metric to a version that operates on the joint state and action space of agents in MARL. We define this metric in the same way as SACo: the ratio

---

2. This setup is reminiscent of *Anscombe's Quartet* (Anscombe, 1973), comprising four datasets that have almost identical summary statistics, yet look markedly different when visualised.

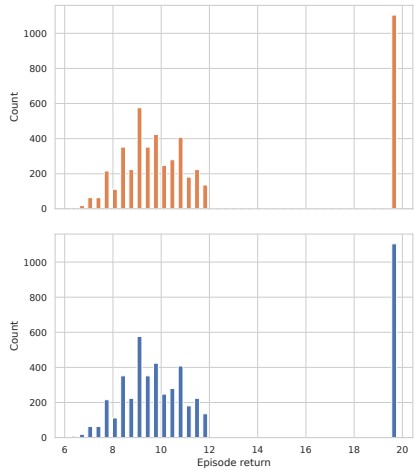

(a) Histograms of the subsampled
`5m_vs_6m` datasets.

(b) Means and standard deviations of the episode
returns in subsampled datasets.

| Dataset | Mean | Stddev | # Traj | # Trans |
|---|---|---|---|---|
| ● CFCQL | 12.05 | 4.36 | 4992 | 140073 |
| ● OG-MARL | 12.05 | 4.36 | 4992 | 134985 |

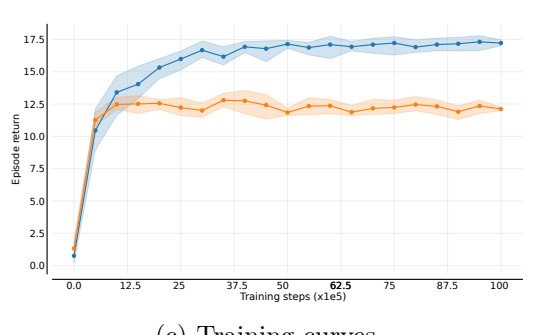

(c) Training curves

Figure 4: We use two subsampled datasets of the `5m_vs_6m` scenario from SMACv1, with almost identical distributions, but from two different sources (Formanek et al., 2023a; Shao et al., 2023a) (the `Medium` quality in both cases). We then train `IQL+CQL` on each dataset and report its final performance. We repeat the experiment over 10 random seeds.

between the number of unique state-action pairs and the total number of state-action pairs in a dataset. In the multi-agent case, the difference is that the action refers to the joint action of all agents in the system. We use this Joint-SACo metric on the `5m_vs_6m` datasets from Figure 4a, with the results given in Table 2.

Whereas the datasets from this example have almost identical return distributions, we see that they have very different values for state-action coverage. In fact, from Table 2 we observe that 90% of the data from Shao et al. (2023a) are repetitions of previously seen state-action pairs, compared to just 17% in the data from Formanek et al. (2023a). This finding illuminates how complexities in multi-agent dynamics fail to be encapsulated solely in episode return values, and why we must be particularly careful.

## 3.2 Quantifying Correlations between Dataset Properties and Algorithm Performance

Having demonstrated how challenging it is to control for the effect of data on algorithm performance through a series of small-scale experiments, we now conduct an extensive survey of how dataset properties are correlated with the performance of a broad range of state-of-the-art algorithms from the literature. We consider datasets and results on SMAC (Samvelyan et al., 2019). SMAC is the environment that is most consistently used across works in the literature (Formanek et al., 2024). Therefore, as a source from which to draw conclusions, it provides some confidence. In particular, we consider three different sources of SMAC datasets, namely from OG-MARL (15 datasets) (Formanek et al., 2023a), CFCQL (16 datasets) (Shao et al., 2023b) and OMIGA (12 datasets) (Wang et al., 2023c). All of the dataset characteristics

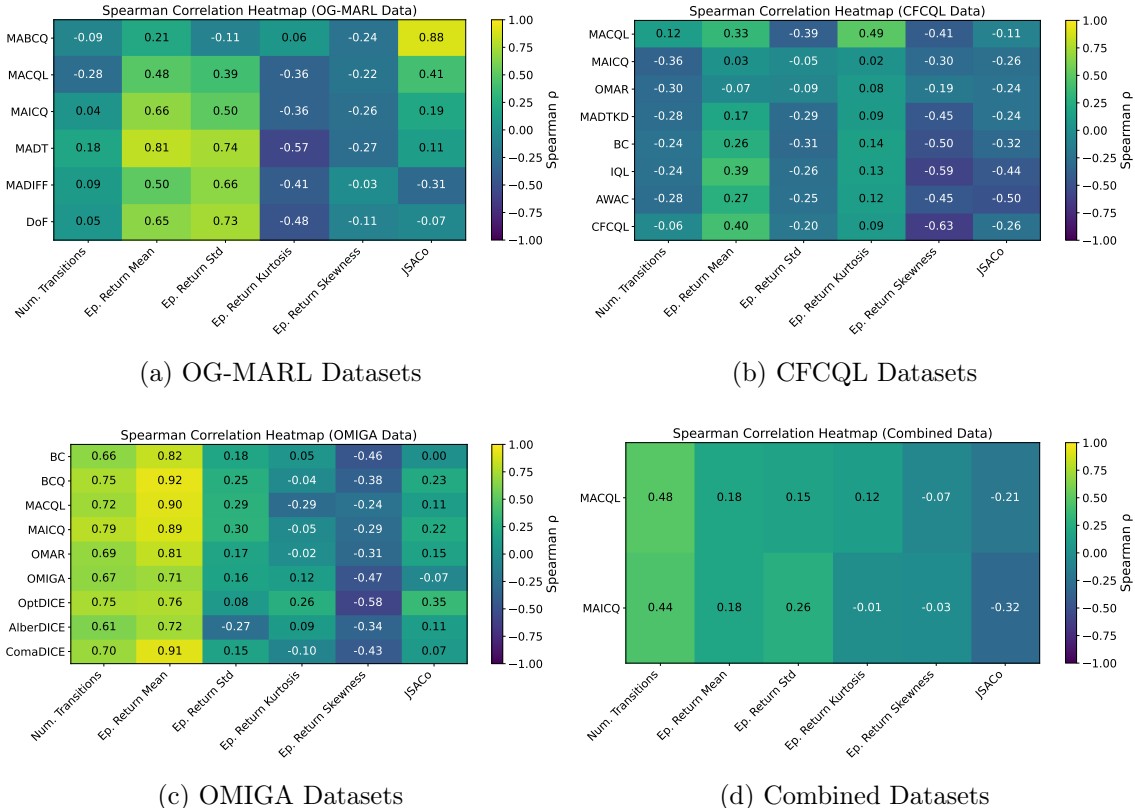

Figure 5: Heatmaps of Spearman correlations between various dataset properties and the reported performance of algorithms across three different collections of datasets, namely OG-MARL (Formanek et al., 2023a), CFCQL (Shao et al., 2023b) and OMIGA (Wang et al., 2023c)). The latest reported state-of-the-art results from the literature were used in each case respectively (Shao et al., 2023b; Li et al., 2025; Bui et al., 2025). In addition, since MACQL and MAICQ are shared across all datasets, we additionally compute correlation scores aggregated across all datasets for these two algorithms.

are documented on dataset cards, available on the OG-MARL website [3]. In addition, we use the latest state-of-the-art results from the literature for each of the datasets. For the OG-MARL datasets, we use the results reported by Li et al. (2025) (6 algorithms), for OMIGA we use the results reported by Bui et al. (2025) (9 algorithms), and for CFCQL we use the original results by the CFCQL authors (Shao et al., 2023b) (8 algorithms). For each dataset source, we plot a heatmap that shows the Spearman correlation between several dataset characteristics discussed in the previous section and the reported performance for each of the algorithms. The correlation scores are computed by aggregating across all of the datasets from the given dataset source. The results of which are given in Figure 5.

**Results.** In OG-MARL, the number of transitions has virtually no effect on performance, as datasets are all approximately the same size, $\sim 1M$ steps by design. In contrast, in the CFCQL

---

3. https://instadeepai.github.io/og-marl/dataset_cards/og_marl/

suite, larger datasets slightly degrade performance for OMAR, MAICQ, MADT+KD, BC, IQL and AWAC (with only MACQL showing a marginal gain). On the OMIGA benchmarks, where dataset sizes vary widely, nearly every algorithm benefits significantly from having access to more transitions. Across OG-MARL, CFCQL and OMIGA, mean episode return is the single most reliable predictor of algorithm performance: higher returns uniformly yield better results. The only exceptions are OMAR and MAICQ in CFCQL, which show negligible correlation. In OG-MARL, greater return variance helps most methods, except for MABCQ. In CFCQL, spread has a small or slightly negative impact across the board. On OMIGA, all algorithms except for AlberDICE benefit modestly from higher variance. OG-MARL datasets have a high average diversity (JSACo aprox. 0.8), with positive correlations between diversity and performance. CFCQL datasets, however, are uniformly low in diversity (JSACo aprox. 0.3), making any diversity effect hard to distinguish. OMIGA datasets all score JSACo near 1, so the impact of diversity cannot be evaluated accurately. The reason OMIGA datasets have such a high JSACo score has to do with the fact that the authors used a variant of SMAC popularised by Yu et al. (2022a), which includes a significant amount of feature engineering, resulting in an observation vector up to 4 times larger than the standard SMAC environment (once again highlighting the challenge of standardisation in the field). Across most datasets and algorithms, skewness was inversely correlated with performance. Highlighting that most existing offline MARL algorithms fail to effectively focus on high-return strategies that occur infrequently in the dataset, which corroborates similar observations made by Tilbury et al. (2024). The effect of kurtosis, on the other hand, was mostly only observable in the OG-MARL data, where the correlation scores were negative. MACQL on the CFCQL datasets and OptiDICE on the OMIGA datasets were the only two cases which showed some positive correlation.

Since MACQL and MAICQ were the only two algorithms which were tested across all datasets, we computed the correlation scores aggregated across all of the datasets for these two alogrithms (see Figure 5d). This gives more reliable correlation estimates for these two algorithms. The size of the datasets is the most correlated with algorithm performance. This comes from the fact that MACQL and MAICQ did poorly on the OMIGA datasets, which were relatively small compared to the OG-MARL datasets, where the algorithms performed significantly better. Next, is the mean and standard deviation of episode return. As one would expect, the mean episode return tends to be positively correlated with performance, although the poor performance on OMIGA datasets once again drew this relationship down. Standard deviation, as the common proxy for diversity, also tends to be positively correlated with performance. However, interestingly, Joint-SACo is negatively correlated. This contrasts with results for single-agent CQL in the work by Schweighofer et al. (2022), where diversity mostly increased performance.

Finally, we consider the interplay between dataset quality and diversity on the performance of offline MARL algorithms, which we show in Figure 6. For MACQL and MAICQ, we plotted each of the reported results on a set of axes, where the X-axis denotes the diversity of the dataset (JSACo) used for training and the Y-axis denotes the dataset quality (Mean Episode Return). The reported performance of the algorithm on the dataset is given by the colour gradient of the marker. The results from the scatter plots corroborate the findings from the correlation heatmaps. Namely, both algorithms clearly benefit from having better-quality data, since many of the light-coloured markers are near the top of the plot. While for diversity,

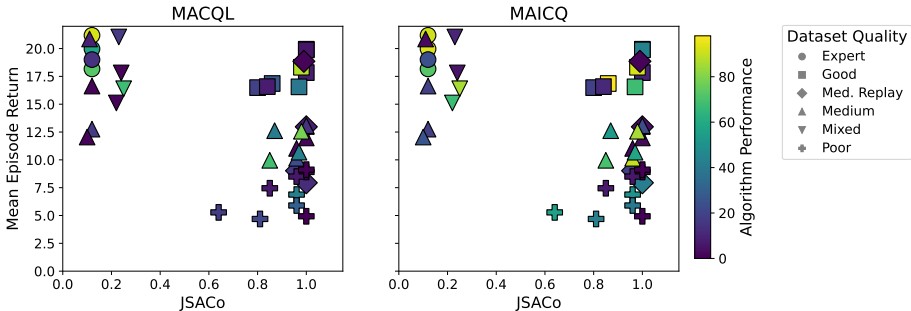

Figure 6: Scatter plot for algorithms MACQL and MAICQ across all datasets from OG-MARL and CFCQL. Each marker represents the reported result of training the algorithm on a given dataset. The x-axis represents the diversity of the dataset (JSACo). While the y-axis represents the quality of the dataset measured in mean episode return. The algorithm performances were scaled between 0-100 across all datasets from the varying sources to allow for aggregation.

the trend is different. Most of the dark markers are on the right, suggesting that too much diversity may be sub-optimal for these two offline MARL algorithms. The distribution of the markers suggests another thing, namely that currently available datasets do not cover the mid-level of diversity (JSACo around 0.5) very well.

### 3.3 Takeaways

Much of the offline multi-agent literature relies on self-generated datasets, frequently accompanied by inconsistent documentation and insufficient consideration for reproducibility. Qualitative descriptors such as "Good," "Medium," or "Mixed" are often used without explicit reference to critical quantitative metrics, such as mean and standard deviation of episode returns. Our systematic exploration of dataset properties highlights several essential insights. Most notably, even datasets sharing identical summary statistics (mean and std) can yield markedly different performances due to underlying distributional differences, emphasizing that these statistics alone are insufficient to fully capture the complexities of multi-agent dynamics. Moreover, two datasets with identical episode return distributions may still lead to significantly different outcomes due to other, less apparent factors such as JSACo, underscoring the critical need for transparent reporting of comprehensive dataset characteristics.

Our extensive correlation analysis spans a broad range of datasets and state-of-the-art algorithms from the literature, providing crucial insights into how different algorithms respond to various dataset characteristics. The analysis highlights that the size of the dataset and diversity metrics such as JSACo significantly impact algorithm performance, sometimes in unintuitive ways. Notably, higher JSACo (indicating greater diversity) may not always correlate positively with performance, highlighting the intricate dynamics unique to multi-agent scenarios. Additionally, skewness in return distributions may also influence algorithm effectiveness, typically negatively correlated with performance, suggesting current offline MARL methods struggle to effectively leverage infrequent high-return experiences.

Ultimately, our analysis stresses an urgent need for increased attention to dataset characteristics in offline MARL research because they are not well understood. Researchers must prioritize transparent, comprehensive documentation and accessible data sharing, including detailed descriptions of generation procedures, distributions, and metrics such as mean return, standard deviation, skewness and JSACo. Moreover, standardizing datasets within the field is essential to ensure consistency, reproducibility, and meaningful comparisons. Beyond these practical measures, our findings pave the way for a promising new research direction investigating precisely how various dataset characteristics influence multi-agent learning, capturing the nuanced interplay of complexity inherent in these environments.

## 4 Putting Data at the Centre of Offline MARL

Having illustrated how a lack of consideration for the impact of data in offline MARL is problematic, we take a step towards alleviating some of the issues we have identified in the previous section. By making three data-driven contributions to the community, we hope to bring data closer to the centre of research in the field. Our first contribution is a set of clear guidelines for how researchers should approach generating datasets in offline MARL. If there is an important reason for authors to generate their own datasets, there should at least be sound principles to follow. Secondly, we significantly enhance the standardisation of data in the field by converting over 80 datasets from prior works into a consistent format (Toledo et al., 2023), which has an emphasis on speed, ease-of-use, clear documentation, and integration into existing frameworks. We upload these datasets to Hugging Face for reliable access in perpetuity.[4] We recommend using existing OG-MARL (Formanek et al., 2023a) datasets for future research and encourage any new datasets generated by the community to be added to the repository following the standards and formats outlined there. That said, we still converted all the datasets we could get access to from prior work for the sake of continuity and the possibility of comparing with those works. Finally, we present an ever-growing repository of open-source tools that can be used to access, analyse, and edit these standardised datasets, for future research.

### 4.1 Dataset guidelines

The gold standard solution is to standardise all of the existing datasets (which we attempt through OG-MARL) and to use a shared methodology when generating novel datasets. When generating a new dataset, there are certainly basic guiding principles that ought to be followed to ensure good scientific practice. We outline such guidelines in the blue box below.

### 4.2 Standardising existing datasets

The single-agent offline RL community have benefited significantly from the widespread adoption of common datasets such as RL Unplugged (Gulcehre et al., 2020) and D4RL (Fu et al., 2020). The offline MARL field could similarly benefit from the adoption of a common set of benchmark datasets.

Although there are multiple possible starting points for standardisation, we recommend OG-MARL (Formanek et al., 2023a). OG-MARL is solely focused on providing standardised

---

4. https://huggingface.co/datasets/InstaDeepAI/og-marl/

---

### Guidelines for generating new datasets for offline MARL research

**1. Is a new dataset really necessary?**

- Is there an existing dataset in the field you could use instead?
- If a new dataset is required for your research, make sure you document why and how exactly your dataset is different.

**2. Have you documented all of the relevant information regarding how you generated your data?**

- Document which environment you used and how people can access the environment. Make sure to be explicit about the version of the environment used. This is to ensure proper version control for comparisons as later environment versions might be released in the future (in some cases from authors different than the original creators).
- Document relevant high-level environment properties e.g. number of agents, action size, observation size, sparse/dense reward and so on.
- Document how you generated the dataset. For example, which online MARL algorithm collected the experience and how did you sub-sample the data?

**3. Have you included a quantitative analysis of the composition of your data?**

- Report the following summary statistics for your datasets: episode returns min, mean, max and standard deviation, as well as the number of episodes and transitions in your dataset.
- Include plots of the episode return distribution e.g. histograms or violin plots.
- Include a measure of action-space coverage for your dataset e.g. Joint-SACo.

**4. Can other researchers access your datasets? And will they still be able to in a year from now?**

- Make a download link easily accessible. Ensure the link will not expire and that downloads are successful from different regions of the world.
- Consider adding your datasets to a community-driven datasets repository such as OG-MARL (Formanek et al., 2023a).
- Use a dataset format that is widely adopted in the field. Or include sufficient documentation on how to load and use your dataset.
- Include a dataset licence.

---

datasets to the community, rather than hosting datasets that exist only as a by-product of algorithmic research. These datasets have already been cited in multiple works (Formanek et al., 2023b; Zhu et al., 2023; Yuan et al., 2023; Formanek et al., 2024; Tilbury et al., 2024; Putla et al., 2024; Ruhdorfer et al., 2024; Jing et al., 2024). The repository is ever-growing and extendable by the community, and supports a wide range of environments.

We standardise the format of these datasets to the open-source and industry-supported utility of `Vault` (Toledo et al., 2023), because of its focus on speed, clear documentation, and easy accessibility. For future offline MARL research, we strongly recommend using OG-MARL datasets in the Vault format. However, to enable comparisons to past works in the literature, we have additionally converted datasets from other authors into the Vault format. In Table 3, we provide a list of the datasets converted and made available to the community in OG-MARL, hosted on the Hugging Face platform.

Table 3: Complete list of datasets converted into the Vault format (Toledo et al., 2023). Though we strongly recommend using OG-MARL in future research, we convert other datasets for continuity with past research. *Environment sources:* MAMuJoCo (Peng et al., 2021), SMACv1 (Samvelyan et al., 2019), SMACv2 (Ellis et al., 2022), MPE (Lowe et al., 2017), RWARE (Christianos et al., 2020)

| Source | Environment | Scenario | Datasets |
|---|---|---|---|
| Formanek et al. (2023a) | MAMuJoCo | `2halfcheetah` | *good, medium, poor* |
| | | `2ant` | *good, medium, poor* |
| | | `4ant` | *good, medium, poor* |
| | SMACv1 | `2s3z` | *good, medium, poor* |
| | | `3m` | *good, medium, poor* |
| | | `3s5z_vs_3s6z` | *good, medium, poor* |
| | | `5m_vs_6m` | *good, medium, poor* |
| | | `8m` | *good, medium, poor* |
| | SMACv2 | `terran_5_vs_5` | *replay* |
| | | `zerg_5_vs_5` | *replay* |
| Pan et al. (2022) | MAMuJoCo | `2halfcheetah` | *expert, medium-replay, medium, random* |
| | MPE | `simple-spread` | *expert, medium-replay, medium, random* |
| | | `simple-tag` | *expert, medium-replay, medium, random* |
| | | `simple-world` | *expert, medium-replay, medium, random* |
| Shao et al. (2023a) | SMACv1 | `2s3z` | *expert, medium-replay, medium, mixed* |
| | | `3s_vs_5z` | *expert, medium-replay, medium, mixed* |
| | | `5m_vs_6m` | *expert, medium-replay, medium, mixed* |
| | | `6h_vs_8z` | *expert, medium-replay, medium, mixed* |
| Wang et al. (2023b) | SMACv1 | `corridor` | *good, medium, poor* |
| | | `2c_vs_64zg` | *good, medium, poor* |
| | | `5m_vs_6m` | *good, medium, poor* |
| | | `6h_vs_8z` | *good, medium, poor* |
| | MAMuJoCo | `2ant` | *expert, medium-expert, medium-replay, medium* |
| | | `3hopper` | *expert, medium-expert, medium-replay, medium* |
| | | `6halfcheetah` | *expert, medium-expert, medium-replay, medium* |
| Matsunaga et al. (2023) | RWARE | `tiny-2g` | *expert* |
| | | `tiny-4g` | *expert* |
| | | `tiny-6ag` | *expert* |
| | | `small-2ag` | *expert* |
| | | `small-4ag` | *expert* |
| | | `small-6ag` | *expert* |
| | | **Total number of datasets:** | **88** |

## 4.3 Dataset analysis tools

Our final contribution towards improving data awareness in offline MARL is a set of tools which can be used to download, subsample, combine, and analyse datasets. These tools, which live in OG-MARL, can be used on any dataset which conforms to the Vault API. The tools are accompanied by a demonstrative notebook, which explains how to use them and provides enough understanding of the Vault and OG-MARL systems to be able to work on custom tools and workflows. Our set of utilities are outlined below.

We provide a demonstration of our tools using the `2s3z` scenario from SMACv1 (Samvelyan et al., 2019), with the dataset from OG-MARL (Formanek et al., 2023a). We focus on dataset

---

**Dataset utilities**

**Simplified loading of datasets**
- Support for downloading all 88 Vault datasets from OG-MARL.
- Support for downloading a Vault from a user-specified URL.

**Dataset analysis tools**
- Dataset structure: `describe_structure` prints the pytree structure of each dataset in the Vault, and gives the number of transitions and trajectories in each dataset.
- Episode returns: `describe_episode_returns` plots for each dataset in the Vault the histogram and violin plot of its episode returns, and outputs a table containing the episode return mean, standard deviation, minimum, and maximum.
- Coverage properties: `describe_coverage` produces a log-log plot of count frequencies of unique state-action pairs, as well as the Joint-SACo value for each dataset.
- Summary: `descriptive_summary` plots episode return histograms, and outputs a table containing episode return mean, standard deviation, minimum and maximum, Joint-SACo, number of transitions and number of trajectories in each dataset.

**Tools to subsample and combine Vaults**
- Subsample a Vault to within one trajectory's length of a specified number of transitions.
- Combine a list of datasets into one larger dataset.
- Subsample two datasets to have near-identical episode return distributions.
- Subsample a dataset to have a specific episode return distribution.

---

analysis (where we give insights into dataset composition), as well as subsampling and combining tools (which can be used for a variety of reasons: to make datasets smaller so that they use less memory, to create datasets for ablations, and to combine datasets when more training data is required).

**Analysis**  Our analysis tools cover all requirements stipulated in the analysis section of our dataset generation guidelines. We provide four high-level functions to generate various insights for a user-specified selection of datasets in the Vault format. Calling `descriptive_summary` with a provided Vault will generate a summary such as the one illustrated by Figure 7, with both tabular and histogram information returned. Users can further access episode return violin plots by calling `describe_episode_returns`, detailed structural information about the Vault by calling `describe_structure`, and state-action count information by calling `describe_coverage`.

From these outputs, we can now analyse the dataset and notice interesting insights. For example, we can see in Table 7 that the `Poor` dataset contains far fewer trajectories than the `Good` and `Medium` datasets, despite containing a similar number of transitions. On average, the episodes in the `Poor` dataset contain more than 100 transitions, which means that the episodes usually roll out to their full length. We also notice that there is not much of a difference between the Joint-SACo of the datasets, though each dataset's coverage is diverse, with at most 4% being repeated pairs in each case. Looking at the skewness values for the datasets we observe that the Good and Medium datasets are skewed to the right. The Poor dataset on the other hand is more symmetrical. This is corroborated by the density plots

(a) Tabular values returned from the `descriptive_summary` function

| Dataset | Mean | Stddev | Min | Max | Mode | Median | Kurtosis | Range | IQ Range | Skewness | # Transitions | # Trajectories | Joint-SACo |
|---------|------|--------|-----|-----|------|--------|----------|-------|----------|----------|---------------|----------------|------------|
| Good | 18.32 | 2.95 | 0.00 | 21.62 | 20 | 20 | 0.52 | 21.6 | 4.2 | 1.39 | 995 829 | 18 616 | 0.98 |
| Medium | 12.57 | 3.14 | 0.00 | 21.30 | 20 | 12.1 | 0.78 | 21.3 | 3.2 | 0.93 | 996 256 | 18 605 | 0.98 |
| Poor | 6.88 | 2.06 | 0.00 | 13.61 | 6.7 | 6.8 | -0.28 | 13.6 | 2.8 | 0.01 | 996 418 | 9 942 | 0.96 |

(b) Visualisations for `Good` dataset

(c) Visualisations for `Medium` dataset

(d) Visualisations for `Poor` dataset

Figure 7: The results returned when calling `descriptive_summary` on the `2s3z` Vault from OG-MARL (Formanek et al., 2023a)

which show that in the Good dataset, roughly 70% of episodes have a return near 20, which is greater than the mean of 18.3. The Poor dataset is more normally distributed in the density plot, and this is reinforced by the QQ plot, which shows that the episode returns are closely distributed along the diagonal line when compared to the standard normal. However, the very low Kurtosis value of -0.28 means there are few extreme values and the distribution is overall a lot flatter and more spread out than the standard normal. The information gained here can at a glance help a researcher understand a dataset better.

**Subsampling and combining datasets** Our tools also allow researchers to stitch and slice datasets. While it is important to keep datasets standardised, experiments may require datasets which do not yet exist; there may be constrained memory requirements, new experiments, or ablations requiring new datasets. Rather than expecting researchers to create entirely new datasets for such experiments, which might introduce inconsistencies

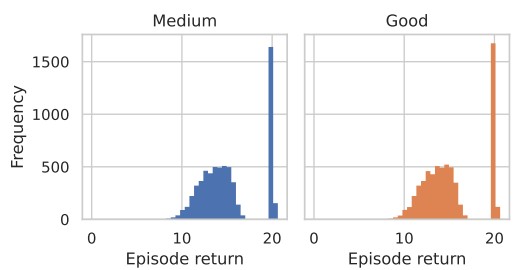

(a) Original `Medium` and `Good` datasets subsampled to have similar episode return distributions.

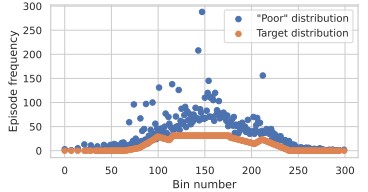

(b) `Poor` dataset distribution versus a target distribution.

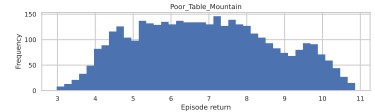

(c) Histogram of the dataset with the desired target distribution, after being subsampled from `Poor`.

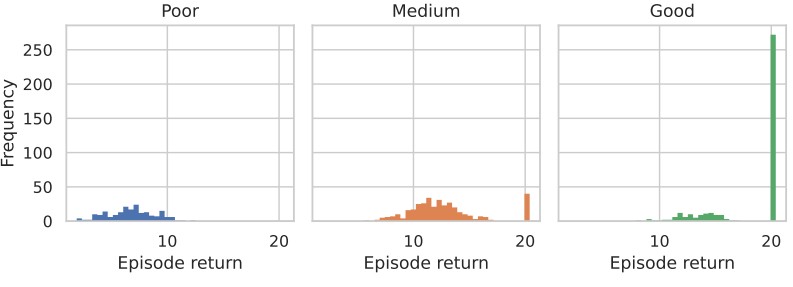

(d) Datasets subsampled to 20 000 transitions each.

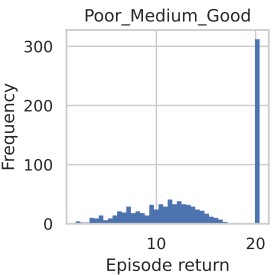

(e) Combined datasets.

Figure 8: Illustrative dataset subsampling examples, using the `2s3z` Vault.

between datasets available in the field, we give researchers the tools to flexibly subsample and combine existing datasets according to the properties that their experiments require. Figure 8 illustrates an assortment of such examples—showing how datasets can be subsampled or combined, and showing how desired distributions can be systematically created. We also provide a notebook in the OG-MARL open-source repository that demonstrates how to subsample a dataset according to a user-specified episode return histogram. Our hope is that even though our subsampling tools allow researchers to manipulate data more easily, our analysis tools make it easy enough to uncover dataset discrepancies that researchers should be dissuaded from changing datasets without disclosing their amendments.

Our work opens up an easy-to-use pipeline for data use, generation and understanding in offline MARL. Researchers can now easily expose the contents of datasets for flexible and well-informed access to the material on which their algorithms train. We hope the impact of our work is better understanding not only of datasets in offline MARL, but also of the environments we use and how a data collecting policy interacts with these.

## 5 Conclusion

In this paper, we have surveyed the literature and found that data is largely neglected in the data-driven field of offline MARL. We have shown why paying attention to data in offline MARL research is crucial, through simple yet illuminating examples. We contribute to the community: a guideline for generating multi-agent datasets; a standardised repository of over 80 `environment-scenario-quality` combinations, with a well-documented and accessible API; and useful tooling to aid the understanding of these datasets. In conjunction, these efforts aim to catalyse progress by aligning the field towards scientific rigour. In doing so, along with other standardisation efforts—e.g. standardised baselines (Formanek et al., 2024)—we feel that the discipline is ripe with opportunity to solve hard problems. We encourage researchers to adopt and extend our offerings, working collectively to push the field forward. By working from a strong foundation together, significant breakthroughs can be made.

## Broader Impact Statement

Our contribution opens the door to an offline MARL field in which careful attention is paid to data. However, the path from our contribution to that goal needs to be taken collectively by the community. If our work becomes widely adopted, then we may be shaping what datasets in offline MARL research look like - standardisation by nature places some restriction on the object being standardised, but our guidelines and contributions take into account that some level of flexibility is also required for progress. It is possible that, using our tools, the field could trend towards either using pre-existing standardised datasets (which is preferable) or again generating their own datasets for new experiments (which will be easy to do using our tooling). It is also true that the subsampling and combining tools which we provide can both improve accessibility and standardisation of datasets but also allow researchers to manipulate data without disclosing their alterations.

We must consider not only research impact, but real-world impact. Our long-term goal for impact is to build solid foundations on which offline MARL can develop, which may accelerate the progress in the field. The potential effects of progress in offline MARL are vast, but our work specifically assists in standardising and analysing datasets. We hope that transparency around the nature of datasets will filter through not only in research but also into real-world scenarios. Dataset transparency in the real world is, however, a far more complex topic since datasets from real-world scenarios may contain sensitive information. Care must therefore be taken in performing and presenting our suggested analyses on real-world datasets.

## Acknowledgments and Disclosure of Funding

This work was supported by InstaDeep Ltd.

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
