# OpenReview forum: "Putting Data at the Centre of Offline Multi-Agent Reinforcement Learning"
_DMLR — Accepted by DMLR_

### Review · Reviewer_W9DT · 2025-01-28

**Recommendation:** 3
**Confidence:** 2

**Summary Of Contributions:**

This paper addresses a critical gap in offline multi-agent reinforcement learning (MARL) by emphasizing the central role of data in experimental design and algorithm evaluation. The authors identify the lack of consistent methodology in dataset generation and limited characterization of datasets in prior research, demonstrating how these issues hinder reproducibility and meaningful comparisons. To address this, the paper makes three key contributions: (1) providing clear guidelines for generating novel datasets, ensuring consistency and transparency; (2) standardizing over 80 existing datasets into a unified repository with a consistent storage format and an easy-to-use API; and (3) developing a suite of analysis tools to facilitate deeper insights into dataset characteristics, supporting both algorithm development and experimental rigor. This work serves as a foundational step towards improving data awareness and experimental standardization in offline MARL, providing researchers with valuable tools and resources to advance the field.

**Strengths:**

1. The paper addresses a critical and often overlooked issue in offline multi-agent reinforcement learning (MARL) — the lack of standardized data usage and its impact on experimental results. This is a highly relevant and impactful topic for the research community.
2. The authors provide a thorough review of existing literature, clearly highlighting the inconsistencies in dataset generation and usage, as well as the insufficient reporting of dataset characteristics in prior work.
3. The paper takes a foundational step toward improving experimental standardization in offline MARL, providing valuable tools and frameworks that can facilitate meaningful comparisons and drive further progress.

**Audience:**

Yes

**Broader Impact Concerns:**

No.

**Claims And Evidence:**

Yes.

**Datasets And Benchmarks:**

Yes.

**Extended Submissions:**

No.

**Limitations:**

1. The dataset characterization relies primarily on simple metrics (e.g., mean, standard deviation, coverage rate) and visualizations (e.g., histograms), which lack depth and fail to fully capture the complexity of the data.
2. The validation of default assumptions is primarily based on visualizations, which introduces subjectivity and weakens the scientific rigor of the conclusions.
3. Summarize high-level principles or findings based on experimental data (e.g., relationships between dataset characteristics and algorithm performance) to provide actionable insights for researchers.

**Requested Changes:**

1. Incorporate additional statistical measures, such as median, mode, range, interquartile range, skewness, kurtosis, minimum, and maximum values. Advanced methods, such as QQ plots and density estimations, could also enhance the analysis.
2. Use statistical hypothesis testing (e.g., t-tests, ANOVA, Mann-Whitney U tests) and confidence intervals to provide robust quantitative support for the conclusions. Consider reporting statistical significance (e.g., p-values) and effect sizes where applicable.
3. The experiments are primarily focused on hypothesis validation, without extracting broader, qualitative rules or insights from the data that could guide future research.

**Strengths And Weaknesses:**

Please see the following Strengths and Limitations.

---

### Review · Reviewer_V8hr · 2025-06-11

**Recommendation:** 3
**Confidence:** 2

**Summary Of Contributions:**

This paper underscores the sensitivity of empirical results in offline MARL research to dataset characteristics. Through a sequence of controlled experiments, the authors demonstrate that variations in dataset properties, such as the mean and standard deviation of episode returns, the finer-grained return distribution, and the joint state-action coverage, can each alter algorithmic performance by large margins. Consequently, without a standardized, well-documented, and easily accessible suite of multi-agent datasets, comparisons between offline MARL algorithms can be confounded by uncontrolled data factors rather than methodological advances.

The authors then provide a checklist for researchers who needs to generate a new dataset:
1) Justification for why a new dataset is needed
2) Documenting environment version, number of agents, observation/action dimensions, behaviour policies used, sampling frequency, etc.;
3) Report min, mean, max, standard deviation, Joint-SACo score, and visualizations of return distributions;
4) Provide persistent download links (e.g., Hugging Face), adopt a common format (Vault), and include license information

The authors also convert convert over 80 existing datasets to the Vault format for consistency, and open-source an analysis and subsampling toolchain to describe the dataset properties and conduct stratified sampling to match a target $\{\mu, \sigma\}$ or a full target histogram.

**Note**: Have raised score 2->3.

**Strengths:**

1) Argument that uncontrolled dataset properties can overturn algorithmic comparisons in offline MARL is clear.
2) The four ablations (across mean, std, shape, joint-coverage) neatly illustrate distinct failure modes.
3) Provide a practical deliverable by way of 88 legacy datasets in a single *Vault* schema, hosted on HuggingFace; plug-and-play loader + subsampling / plotting helpers for ease of downstream research.
4) Provides an actionable best-practice template for future dataset papers.
5) Joint-SACo metric extends state-action coverage to the multi-agent joint space, a simple and easy to compute metric, potentially standardizable.

**Audience:**

Yes

**Broader Impact Concerns:**

Can the authors comment on the maintenance burden with maintaining the Vault hub? Also, future community uploads may include proprietary or privacy-sensitive logs; Vault doesn’t enforce license checks. Should an upload checklist be published, where license type is stated, mandates to scrub personally-identifiable data?

**Claims And Evidence:**

The evidence supports the thesis, but the aforementioned issues with scale (only algorithm is CQL, limited testing of relationship of Joint-SACo with final return, fairly limited agent counts, etc.) limit the inferences able to be drawn.

**Datasets And Benchmarks:**

The authors detail how the used datasets are generated.

**Extended Submissions:**

N/A

**Limitations:**

1) All learning runs rely on CQL-style algorithms; no BC, decision-transformer, or model-based baselines. Sensitivity conclusions may be CQL-specific.
2)  σ-ablation confound – episode count fixed, but # transitions / Joint-SACo not reported. Sets with larger σ likely contain more unique states/diverse trajectories. Performance differences may stem from dataset volume, not spread alone. While the paper notes that the relationship between σ and performance is non-monotonic, as it stands we cannot disentangle a) effect of return spread, n) effect of sheer data volume, c) effect of coverage.
3) Joint-SACo not validated at scale – only shown for a single two-dataset comparison.
4) Vault `metadata.json` appears to omit return histograms, σ, Joint-SACo, behaviour-policy tags. Users still have to recompute key stats; undercuts “standardization” goal.
5) For a claim that dataset properties broadly dictate offline-MARL outcomes, one would perhaps expect more domains and/or higher agent counts. Higher agent counts stress-test joint coverage claims: the joint action space grows combinatorially with the number of agents, and so achieving the same Joint-SACo with more agents requires magnitudes more transitions.
6) Conversion to Vault + helper scripts, while useful, are largely infrastructural; scientific contribution relies mainly on small-scale ablations. Even as an engineering contribution, they can be quickly implemented by a standalone researcher.

**Requested Changes:**

1) Re-run (or augment) the “Dataset Spread” study so that all five σ-bands have exactly the same number of transitions and identical Joint-SACo (or report the residual differences).

2) Compute Joint-SACo for all 80 + Vault datasets and report its correlation (Pearson/Spearman) with final return for a fixed offline agent; include a scatter plot in the appendix.

3) Add behavioural cloning and/or decision transformer, etc. as another algorithm, to demonstrate data-sensitivity patterns are not exclusive to CQL.

4) Embed in metadata.json for every dataset:
• 𝜇 , 𝜎, min, max,
• return histogram (binned)
• Joint-SACo
• # episodes, # transitions
• behaviour-policy description
• environment version/commit.

5) Rather than stating "we fix the standard deviation at approximately 2.0 as calculated over 2000 episodes", list the actual standard deviations. Also label all figures with the values of the mean, std, etc., so the figures are self-contained.

6) Addition of larger MuJoCo and SMAC maps?

**Strengths And Weaknesses:**

**Strengths**: The thesis, that uncontrolled dataset properties can overturn algorithm comparisons, is clear and intuitive. The authors provide a progressive demo by way of 4 ablations to illustrate distinct failure modes. The authors also introduce a new metric (Joint-SACo) that extends state-action coverage to the multi-agent joint space. See Strengths section for more detail.

**Weaknesses**: The present experiments are somewhat narrow. The ablations at present I believe have some significant confounding factors that limit the inferences we can draw. See Limitations for more detail.

---

### Review · Reviewer_b7Jg · 2025-10-16

**Recommendation:** 4
**Confidence:** 3

**Summary Of Contributions:**

The authors identify and address a clear gap in the offline MARL literature, namely the neglected role of datasets in driving performances. The paper first surveys prior work to show that most offline MARL publications create and use self-generated datasets with inconsistent documentation and sparse reporting of dataset properties (Sec. 2). Building on this diagnosis, the authors present empirical evidence demonstrating that dataset characteristics, such as mean episode return, return spread, underlying distributional shape, and joint state-action coverage, can affect the outcomes of offline MARL algorithms (Sec. 3.1). In response, the paper provides three community-facing contributions: a set of practical guidelines for generating and reporting datasets (Sec. 4.1), a standardisation effort that converts over 80 datasets into a common Vault format and hosts them (Sec. 4.2), and an accompanying suite of analysis and subsampling tools (Sec. 4.3). These contributions are supported by illustrative experiments mainly on SMAC and MAMuJoCo benchmarks, correlation analyses linking dataset metrics to algorithm performance (Sec. 3.2), and a public repository to facilitate adoption.

**Strengths:**

**Significance**

 The work addresses a central problem, as Offline MARL's lack of standardisation around datasets threatens the interpretability and reproducibility of empirical claims. By focusing on datasets, the paper drives the attention away from isolated algorithmic tweaks towards a more rigorous approach. The significance lies less in proposing a new learning algorithm and more in providing the scaffolding (datasets, tools, and reporting practices) that allows the community to make meaningful, verifiable progress.

**Relation to prior work**

The authors/ survey on how prior papers report (or fail to report) dataset properties is convincing and aligns with concerns raised in related single-agent studies about dataset-driven evaluation. The extension of State–Action Coverage to a joint multi-agent Joint-SACo metric helps reveal nuances that episode-return summaries miss. The work also offers a common empirical substrate for fair comparisons.

**Relevance to the broader research community**

 This paper will be of interest to practitioners and researchers who develop offline MARL algorithms, who publish empirical results in the area, and who is concerned with reproducibility and benchmarking. The practical repository and API hosting, together with the notebooks and subsampling utilities helps reproducible experiments and (hopefully) increases adoption in both academia and industry.

**Quality of the research**

 The experiments are thoughtfully constructed and the paper demonstrates a clear expertise of relevant baselines and prior datasets.  The correlation analysis across multiple dataset collections and algorithms provides a valuable high-level map of how dataset metrics associate with algorithm performance.  The paper acknowledges several of these limitations in its takeaways (Sec. 3.3), which is good scientific practice.

**Clarity of the paper**

 Overall, the manuscript reads clearly. The narrative is logical and well paced. Figures and tables support the arguments effectively.

**Ethical and social implications**  I believe the Broader Impact statement sufficiently covers possible scenarios, noting that dataset standardisation can both improve reproducibility and introduce risks, such as easier undisclosed manipulation via subsampling tools and handling of sensitive real-world data. The recommendation to include dataset licenses, use stable hosting, and document provenance is appropriate.

**Audience:**

Yes

**Claims And Evidence:**

Yes

**Datasets And Benchmarks:**

Yes

**Extended Submissions:**

N.A.

**Limitations:**

Just as an exercise of robustness, it would be interesting to add some empirical evidences past cooperative SMAC scenarios and MAMuJoCo tasks, to test whether these claims transfer. Again, this would be more of a nice-to-have rather than an actual limitation.

**Requested Changes:**

I would strongly suggest to include a formal definition of Joint-SACo, as it would allow to both have a stronger intuition of the metric and justify the conclusions.

**Strengths And Weaknesses:**

**Strenghts**

The manuscript’s main strength is its data-centric framing applied to a field that is data-driven but has not treated datasets systematically:  the empirical demonstrations in Sec. 3.1 are well chosen to expose failure modes that simple statistics miss. On the other hand, coverage (Joint-SACo) and distribution shape do matter; and dataset size and diversity interact with offline algorithms in nontrivial ways. In term of prescriptive contributions, by converting a broad collection of prior datasets into a standard format and providing tooling for analysis and subsampling, the authors deliver immediate practical value that lowers the barrier for reproducible research and more careful benchmarking. The guidelines for dataset generation and reporting (Sec. 4.1) are concise and actionable, and the inclusion of notebooks and Hugging Face hosting increases the likelihood of community uptake.

**Weaknesses**

There are some weaknesses and limitations:

-  The empirical scope focuses primarily on cooperative SMAC scenarios and MAMuJoCo tasks (Sec. 3). Thus, generalisation to other multi-agent settings is suggested but not demonstrated. Having said that, most of the offline MARL literature focuses on these tasks only, so I believe that adding another out-of-sample task would be just a nice to have.
-  Some of the analyses, including the heatmaps and scatterplots, are mostly observational and may conflate confounded variables (for example, dataset generation protocols, feature engineering choices across dataset sources or algorithm hyperparameters). A deeper controlled study, beyond subsampling and matching by episode return statistics, would strengthen the claims about which dataset properties truly drive algorithmic performance. Again, this would be a nice to have in my view.